# Profiling of the Citrus Leaf Endophytic Mycobiota Reveals Abundant Pathogen-Related Fungal Groups

**DOI:** 10.3390/jof10090596

**Published:** 2024-08-23

**Authors:** Feng Huang, Jinfeng Ling, Yiping Cui, Bin Guo, Xiaobing Song

**Affiliations:** Plant Protection Research Institute, Guangdong Academy of Agricultural Sciences, Key Laboratory of Green Prevention and Control on Fruits and Vegetables in South China Ministry of Agriculture and Rural Affairs, Guangdong Provincial Key Laboratory of High Technology for Plant Protection, Guangzhou 510640, China; rm12407@126.com (F.H.);

**Keywords:** endophyte, fungal community, citrus, pathogen, mycobiota

## Abstract

Plant endophytic microbial communities consist of many latent plant pathogens and, also, many pathogen-related species with reduced virulence. Though with a long history of co-evolution, the diversity and composition of the endophytic mycobiota, especially the pathogen-related fungal groups, has been under-investigated in *Citrus* (*C.*). Based on the amplicon sequencing of fungal internal transcribed spacer (ITS), the leaf endophytic mycobiota were profiled on citrus varieties from different citrus-producing regions. The pomelo variety shared significantly distinctive leaf mycobiota when compared to the mandarin and sweet orange; these conform to their host genetic relationships. In addition, a data set of 241 citrus-related fungi, including 171 (71%) pathogens and potential pathogens, was summarized from previous studies. Under the criteria of local BLAST (covered ITS nucleotide ≥ 150 bp, sequence identity ≥ 99%), a total of 935 fungal operational taxonomic units (OTUs) were assigned to 62 pathogen-related fungal groups, representing 14.9% of the relative abundance in the whole community. Of which, the top groups consisted of *Colletotrichum gloeosporioides* (mean relative abundance, 4.3%), *Co. citricola* and *Co. karstii* (2.7%), *Zasmidium citri-griseum* (2.4%), and *Z. fructigenum* (1.4%). At the genus level, the ratio of the pathogen-related fungal groups in 64% of fungal genera (16 out of 25) exceeded 50%, which are the solely or mainly occurring fungi of their genus in citrus. Our study suggests that the leaf endophytic compartment may be an important place for the growth of latent pathogens.

## 1. Introduction

Citrus is one of the most important fruit crops globally, with nearly 140 producing countries across the temperate and tropical regions [1]. In China, citrus is cultivated in about 20 provinces, autonomous regions, and municipalities; the total cultivating area is up to 2.5 million ha [2]. The most widely cultivated citrus species are mandarin (*C. reticulata*), sweet orange (*C. sinensis*), pomelo (*C. grandis*), grapefruit (*C. paradisi*), and lemon (*C. limon*), among which mandarin consists of nearly two-thirds of all the citrus production in China [3]. Based on partial or whole genome sequences, sweet orange and mandarin share a closer phylogenetic relationship when compared to pomelo, grapefruit, and lemon [4,5]. The differences in phylogeny may influence the occurrence of fungal pathogens and endophytic mycobiota of citrus varieties.

During citrus growth, tens of fungal diseases can occur from the plant rhizosphere (root and ambient soil) to the phyllosphere (trunk, branch, twig, leaf, flower, and fruit). For many of these diseases, one symptom can be caused by several closely-related fungi [6,7,8,9], which leads to a high diversity of citrus fungal pathogens. For example, the anthracnose spot on citrus leaves and fruit can be caused by nearly 30 species of the genus *Colletotrichum* [8,10,11], and the decay of citrus trunk, branch, and twig has been associated with at least 26 species of 8 genera in the family *Botryospaeriaceae* around the world [6,12,13]. For the last decade, the development of next generation sequencing technologies has largely expanded the number of fungi that we can identify from a single plant [14]. For example, a mycobiota from a single plant leaf may consist of hundreds and thousands of fungi [15,16]. With so many co-occurring fungi, it is very challenging to understand the function of each one of them, as well as their function as a whole community [17,18,19,20].

After profiling the microbial community, many studies have tried to culture as many members as possible in a bacterial or fungal community and analyze the function (plant growth promoting, plant pathogenicity, etc.) of each species [21,22]. This bottom-up method can produce a reference database of bacteria or fungi and facilitate related studies on the same host. However, it is laborious; such work has only been done on the research of mammals and several model plants [21,23,24]. Due to global research on citrus diseases, citrus-related fungi have been well-classified, especially the species of the genera *Colletotrichum*, *Diaporthe*, *Phyllosticta*, and *Zasmidium*. Their attributes (lifestyle, distribution, host specificity, reference sequence, etc.) have also been studied [6,7,8,9,11,13,25,26]. We suggest that an overall summary of the reported citrus-related fungi should be meaningful at this time; a compiled data set, including the information of fungal species, available barcoding gene, lifestyle, pathogenicity, etc. can definitely facilitate the identification, trace, and study of dynamics of the different fungal groups on citrus.

According to Hardoim et al. (2015), plant fungal endophytes are defined by their colonization niche, but not by their interactions with the host plant. The types of interactions between plant and the fungal endophytes range from mutualism to pathogenicity [27,28]. Pathogenicity is a complex phenomenon which can be affected by multiple factors such as the virulence of pathogen, host genotypes, environmental stresses, and microbial interactions [27,29,30]. Therefore, many pathogens can live within plant tissues without causing any symptoms [27,28]. For example, the pathogen *Fusarium verticillioides* in maize, and *Verticillium dahlia* in strawberry, potato, and olive can both live asymptomatically within their host plant [31,32]. Even though the endophytic fungal community acts as an important reservoir of plant pathogens, to our knowledge, limited studies have focused on the composition and distribution of the pathogen-related fungal groups in the endophytic community.

In this study, we collected leaf samples from lemon, mandarin, sweet orange, and pomelo in China and profiled their endophytic fungal communities using amplicon sequencing. We demonstrated that the diversity and composition of the endophytic fungal communities of pomelo leaves was significantly different to those of mandarin and sweet orange, which corresponded to the phylogenetic relationships of the hosts [4,5]. At the same time, we summarized 241 fungi that have been reported mainly as on citrus pathogens or potential citrus pathogens and retrieved their sequence of internal transcribed spacers (ITSs) to build a reference data set. By annotating the sequences with our summarized reference data set, we analyzed the occurrence and distribution of pathogen-related fungal groups in the endophytic fungal community of the citrus leaf and quantified the prevalence of these fungal groups in their genus.

## 2. Methods

### 2.1. Summary of Citrus-Related Fungi

The fungal diseases and fungal species reported on citrus were mainly summarized from three databases: (1) Fungal database of US national fungus collections (https://nt.ars-grin.gov/fungaldatabases/, accessed on 21 April 2022), (2) Common names of plant diseases compiled by the American Phytopathological Society (https://www.apsnet.org/edcenter/resources/commonnames/Pages/default.aspx, accessed on 21 April 2022), and (3) Common names of plant diseases in Japan compiled by the Phytopathological Society of Japan (https://www.gene.affrc.go.jp/databases_en.php, accessed on 11 November 2021). The current legal fungal names were checked according to the records in the database Index Fungorum (http://www.indexfungorum.org/names/Names.asp, accessed on 21 April 2022). The availability of the sequence of the nuclear ribosomal internal transcribed spacer (ITS rDNA) region was checked in the publications for each fungal species. If available, their ITS sequence was retrieved from the nucleotide database of the National Center for Biotechnology Information (NCBI, https://www.ncbi.nlm.nih.gov/, accessed on 22 April 2022) through their accession number. All the ITS sequences were prepared to form our own data set of citrus-related fungal sequences by using BLAST+ software (version 2.12.0, https://ftp.ncbi.nlm.nih.gov/blast/executables/blast+/LATEST/, accessed on 23 April 2022).

### 2.2. Sampling of Citrus Leaves

Leaf samples were collected in China from different citrus varieties: cultivated lemon, mandarin and tangerine, sweet orange, and pomelo. The sampling orchards were widely distributed in different provinces, autonomous regions, and municipalities, such as Chongqing, Guangdong, Guangxi, Guizhou, Hubei, Jiangxi, Sichuan, and Zhejiang (Table 1). In each orchard, one to six citrus trees of the same variety were chosen, and one twig with more than five leaves was cut from each tree to form a leaf sample. The collected samples were kept on ice and sent to the lab as soon as possible. One healthy leaf, without any visible symptoms, was picked from each twig for microbial DNA extraction and amplicon sequencing.

### 2.3. Molecular Analysis of Leaf Endophytic Fungi

To remove leaf epiphytes, each leaf was surface-sterilized by soaking in 1 × TE buffer (supplemented with 0.1% Triton X-100) for 30 s, 75% ethanol for 15 s, 2% bleach for 15 s, and rinsing 3 times in sterilized water. Then, the leaf was dried with a sterilized filter paper and put into a 50 mL tube. All leaf samples were kept in a −80 °C refrigerator for long-term use. The total DNA of the microbes in the leaf was extracted by using the CTAB (cetyltrimethylammonium bromide) protocol [33]. DNA concentration and purity was monitored on 1% agarose gels, and then DNA was diluted to 1 ng/µL with sterile water. The partial nucleotide sequence of the nuclear ribosomal internal transcribed spacers (ITS rDNA) was amplified by polymerase chain reaction (PCR) with fungal primer set ITS1-1F (5′-CTTGGTCATTTAGAGGAAGTAA-3′) and ITS1-1R (5′-GCTGCGTTCTTCATCGATGC-3′). The PCRs were carried out with 15 µL of Phusion^®^ High-Fidelity PCR Master Mix (New England Biolabs, Ipswich, MA, USA), 2 µM of forward and reverse primers, and about 10 ng of template DNA. The thermal cycling program was set as follows: initial denaturation at 98 °C for 1 min, followed by 30 cycles of denaturation at 98 °C for 10 s, primer annealing at 50 °C for 30 s, extension at 72 °C for 30 s, and a final extension at 72 °C for 5 min. For each leaf sample, the PCR was conducted in triplicate and the PCR products were pooled to make one PCR mixture. The mixture of PCR products was viewed by electrophoresis on 2% agarose gel, the target DNA band was selected and purified with Qiagen Gel Extraction Kit (Qiagen, Hilden, Germany). The library for amplicon sequencing was generated by using the TruSeq^®^ DNA PCR-Free Sample Preparation Kit (Illumina, San Diego, CA, USA) and then sequenced on an Illumina NovaSeq platform (Novogene, Beijing, China).

Raw 250 bp paired-end reads were merged by using FLASH (version 1.2.7, http://ccb.jhu.edu/software/FLASH/, accessed on 19 April 2021). By following the quality control process of QIIME (version 1.9.1, http://qiime.org/, accessed on 19 April 2021), reads were subsequently denoised and quality filtered by removing low-quality reads with eight or more homopolymer bases, ambiguous base calls and/or average quality < 25 bases. Quality reads were assigned to each sample according to their unique barcodes. Chimeric sequences were identified and removed from each sample using UCHIME (version 4.1, http://www.drive5.com/usearch/manual/uchime_algo.html, accessed on 19 April 2021). Operational taxonomic units (OTUs) were clustered at the 99% similarity level with the software UPARSE (version 7.0.1001, http://drive5.com/uparse/, accessed on 19 April 2021). To assign taxonomical information to each OTU, the representative sequences from each OTU were blasted against the UNITE ITS sequence database (https://unite.ut.ee/, accessed on 19 April 2021). The pathogen-related fungal OTUs were identified by blasting the summarized citrus-related fungal ITS sequences with the criteria as follows: covered nucleotide ≥ 150 bp, sequence identity ≥ 99%.

### 2.4. Phylogenetic Analysis

Sequences were aligned with the MUSCLE method implemented in MEGA (version 7.0.26, https://megasoftware.net/, accessed on 4 June 2021). The aligned sequences were loaded into QIIME (version 1.9.1) to calculate a phylogenetic tree with the FastTree method [34]. The resultant phylogenetic tree was submitted to the EvolView (version 3) webserver [35] for visualization and annotation.

### 2.5. Statistical Analysis

The OTU table for all samples was generated by QIIME (version 1.9.1), the alpha diversity indices (Observed species, Shannon, Simpson, ACE, Chao1, Good’s coverage, and Phylogenetic diversity) and the relative abundance of each taxon were subsequently calculated in the same software of QIIME. All statistical analyses were conducted in the R software (version 4.0.0, http://www.r-project.org/, accessed on 4 June 2021). To compare the differences of alpha diversity indices and the relative abundance of specific taxa among samples from four citrus varieties, analysis of variance (ANOVA) with Tuckey test was used. The beta dispersion was calculated with a betadisper function in the vegan package in R, which is a multivariate analogue of Levene’s test for homogeneity of variances. After that, PERMANOVA was used to compare the beta diversity using the adonis function in the vegan package. The levels of significance were set at *p* < 0.05 and *p* < 0.01.

## 3. Results

### 3.1. A Summary of Reported Citrus-Related Fungi

In total, 241 fungal ITS sequences were retrieved from the NCBI GenBank database (Figure 1). These sequences were all reported on citrus but with different lifestyles, 171 sequences (71%) were from plant pathogens or potential pathogens, 27 sequences (11.2%) were from either endophytes or epiphytes, and 43 sequences (17.8%) were ambiguous and could not be assigned to either pathogen, endophyte, or epiphyte. Taxonomically, 105 (43.6%) and 84 (34.9%) sequences were from the class *Sordariomycetes* and *Dothideomycetes* in *Ascomycota*, respectively, representing the classes with the most diverse pathogens.

Among all fungi, eleven fungal groups (165 species, 68.5%) at either the family or genus level were singled out based on: (1) their phylogenetic relatedness, (2) seriousness of caused diseases, and (3) similarity of caused symptoms on citrus (Table 2). The group Alternaria (5, 2.1%), Cladosporium (5, 2.1%), Colletotrichum (36, 14.9%), Diaporthe (30, 12.4%), Elsinoë (3, 1.2%), Mycosphaerellaceae (18, 7.5%), and Phyllosticta (8, 3.3%) consisted of fungal species that can infect citrus leaves and mainly occur on the phyllosphere parts of citrus plants. The group Penicillium and Aspergillus (6, 2.5%) consisted of fungal pathogens causing fruit mold during storage. The group Botryospaeriaceae (26, 10.8%), Fusarium (17, 7.1%), and Phytophthora (11, 4.6%) consisted of fungal pathogens that mainly infect citrus trunk, branch, and twig; the latter two also consisted of fungal pathogens that colonize the rhizosphere soils and citrus roots.

### 3.2. Diversity and Composition of Citrus Leaf Endophytic Fungal Communities

A total of 161 leaf samples were collected from four varieties of cultivated citrus and eight citrus producing regions in China (Table 1). Specifically, seven samples were collected from lemon, 86 samples from mandarin, 41 samples from sweet orange, and 27 samples from pomelo. After quality filtering, 3,044,957 sequences were harvested with an average of 18,913 sequences per sample. The rarefaction curves of the observed species indicated that the number of detected fungal OTUs of our samples reached the plateau phase (Figure 2A).

From the results, citrus variety significantly affected the alpha and beta diversity of leaf endophytic fungal communities. The alpha diversity was represented by Observed species (*p* = 0.28, Figure 2B) and Shannon index (*p* < 0.01, Figure 2C); the significant differences of community composition among varieties were tested at the OTU level (*p* < 0.01, Figure 2D). These differences were mainly caused by the pomelo leaves, the average Shannon diversity of pomelo leaves (means ± SD, 4.9 ± 0.8) was significantly higher than those of sweet orange (3.4 ± 1.7, *p* < 0.01), and higher than lemon (3.9 ± 1.5) and mandarin (4.1 ± 1.6) on average. In addition, the fungal composition and structure of pomelo leaves was significantly distinctive compared to mandarin (*p* < 0.01) and sweet orange (*p* < 0.01) leaves (Figure 2D). At the genus level, the composition of the citrus endophytic fungal community was typically represented by only a few genera (Figure 2E), with the genera *Russula* (the average relative abundance of all samples, 25.7%), *Colletotrichum* (9.9%), an unidentified genus of *Cladosporiaceae* (7.7%), and *Zasmidium* (7.4%) leading the rank of relative abundance. Compared to mandarin and sweet orange, the pomelo leaves showed a significantly reduced relative abundance of *Russula* (28.7% and 37.9% to 2%, both *p* < 0.01) and a significantly increased relative abundance of *Zasmidium* (7.7% and 0.8% to 18.4%, *p* = 0.02 and *p* < 0.01, respectively).

### 3.3. Distribution of Pathogens and Potential Pathogens in Citrus Leaf Endophytic Fungal Communities

After blasting the sequences of citrus-related fungi, 935 out of 6667 (14%) OTUs were assigned to 62 fungal groups (Table 3). For example, one OTU (OTU186) matched to 6 species (*L. iraniensis*, *L. mediterranea*, *L. parva*, *L. pseudotheobromae*, *L. subglobosa*, and *L. theobromae*) of the genus *Lasiodiplodia* at the same time, so it was assigned to a complex fungal group (Lasiodiplodia iraniensis.mediterranea.parva.pseudotheobromae.subglobosa.theobromae). For the genus *Zasmidium*, 40, 20, and 130 OTUs were assigned to the species *Z. citri-griseum*, *Z. fructicola*, and *Z. fructigenum*, respectively. Accordingly, these OTUs were assigned to the Z. citri-griseum group, Z. fructicola group, and Z. fructigenum group, respectively. Amazingly, up to 384 OTUs were assigned to multiple species from the *Colletotrichum gloeosporioides* species complex, all these OTUs were assigned to the Co. gloeosporioides group.

These 62 fungal groups, consisting of 14.9% of the relative abundance of the whole fungal community, were further assigned to the eleven fungal pathogen groups. Except for the pathogen groups Elsinoë and Phytophthora, all other nine pathogen groups were found to have closely-related fungi living as endophytes in citrus leaves (Figure 3A). Of these, the Co. gloeosporioides group (average relative abundance, 4.3%), the Co. citricola/karstii group (2.7%), the Z. citri-griseum group (2.4%), and the Z. fructigenum group (1.4%) shared the highest relative abundances (Figure 3B,C).

### 3.4. Dominance of Latent Pathogens within Their Genus

Furthermore, the OTUs assigned to pathogens and potential pathogens were summarized again at the genus level, which resulted in 25 out of 29 genera (excluded were *Acrodontium*, *Coniella*, *Epicoccum*, and *Stemphylium*) consisting of pathogen-related fungal OTUs (Figure 4A). The total relative abundance of the pathogen-related fungal OTUs was listed for each genus (Figure 4B), and the relative abundance of each genus was listed alongside (Figure 4C). The ratio of the pathogen-related fungal OTUs was calculated as the percentage of pathogen-related fungal OTUs in each genus (Figure 4D). Based on these, we found that most of the pathogen-related fungal OTUs showed a superiority in colonizing the citrus leaf endophytic space compared to their peers within a genus. Specifically, the ratio of the pathogen-related OTUs in 64% genera (16 out of 25) exceeded 50%. For some important genera: the number was 93.8% for *Colletotrichum* (pathogen-related OTUs versus others, *p* < 0.01), 58.7% for *Zasmidium* (*p* < 0.05), 95.2% for *Alternaria* (*p* < 0.01), 94.4% for *Pallidocercospora* (*p* < 0.01), and 55% for *Diaporthe* (*p* < 0.05, Figure 4D).

## 4. Discussion

### 4.1. The Composition of the Leaf Endophytic Fungal Community Correlates to the Phylogenetic Relationship of the Citrus

Host factors, such as plant genotype, ecotype, and plant age, are among the major determinants in structuring plant microbial communities [67,68]. In this study, we focused on the leaf endophytic mycobiota of four varieties of cultivated citrus in China (Table 1). The most samples (86 samples, 53.4%) were collected from mandarins, followed by from sweet oranges (41, 25.5%), pomelos (27, 16.8%), and lemons (7, 4.3%). The ratios of collected leaf samples of different citrus varieties were roughly in accordance with their distribution in China. For example, mandarin consisted of nearly 2/3 of all the citrus production, while lemon is not commonly cultivated in China [3].

Phylogenetically, lemon and pomelo are outer species compared to mandarin and sweet orange, the latter two are more closely-related [4,5]. In accordance with this, the leaf endophytic fungal communities of pomelo were significantly distinctive compared to those of mandarin and sweet orange. This difference was mainly reflected in the relative abundances of the species of *Russula* and *Zasmidium*, which suggests that host factors play a role in recruiting specific fungal taxa. The species of *Zasmidium* are probably potential pathogens living as endophytes [25]; the species of *Russula* may offer some benefits to their host plant [69,70].

### 4.2. A Summarized ITS Data Set Is Useful in Studying Citrus Mycobiota

ITS is a universal barcoding gene for fungal phylogenetic analysis [71], and also has been broadly used for the profiling of fungal communities colonizing different ecological niches [72,73]. In searching against fungal databases, such as UNITE and the RefSeq Targeted Locus (Loci) database of NCBI, ITS offers limited information that may assign its origin to many different fungal species of different hosts and geological locations. Works on *Arabidopsis* have tried to isolate and identify a whole collection of microbes occurring on *Arabidopsis* plants, which largely facilitates the identification and trace of the microbial species from the same host plant [23,24,74]. Due to the contribution of many researchers on citrus-related fungi, here we are proposing and testing a data set of 241 fungal ITS sequences to be used on citrus-related studies.

The trace of different pathogens can reflect their temporal dynamics in orchards, thus being meaningful for the study of disease epidemiology [15,75]. From our study, the closely-related fungal groups of several important pathogens, such as *D. citri*, *P. citriasiana*, *Penicillium digitatum*, three species of *Zasmidium*, and *Co. Gloeosporioides*, could be traced based on a partial ITS sequence. To improve the trace accuracy, quantitative PCR methods should be applied along with the amplicon sequencing method. However, tracing the pathogen-related groups may also be useful in practice, because species from the same group may all be pathogenic to citrus. For example, *Co. citricola* and *Co. karstii*, or different species in the *Co. gloeosporioides* species complex generally cause anthracnose symptoms on citrus [8,11,42,43]; *P. citricarpa* and *P. paracitricarpa* both cause citrus black spot [7,61,62,63], *P. capitalensis* and *P. paracapitalensis* both are common endophytes or epiphytes on citrus [7]. However, to further improve the usage of our data set, more fungi should be isolated and identified from citrus, and also, their pathogenicity and other functions to citrus should also be tested.

### 4.3. Leaf Endophytic Community May Be a Reservoir for Citrus Pathogens

During the last decade, citrus anthracnose, caused by *Colletotrichum* spp., citrus greasy spot, caused by *Mycosphaerellaceae* species, Alternaria brown spot, caused by *Alternaria* spp., Melanose, caused by *Diaporthe citri*, and citrus black spot, caused by *Phyllosticta* spp., have been the most influential fungal diseases on citrus in China [45,62]. Accordingly, the fungi from these pathogen-related groups were commonly found in citrus leaf endophytic communities, especially the ones from the groups of Colletotrichum and Mycosphaerellaceae (Figure 3 and Figure 4). Several species, like *Co. gloeosporioides*, *Co. fructicola*, *D. citri*, and *P. capitalensis* (a global endophyte on citrus), have previously been isolated from asymptomatic citrus tissues [10,26,63], and the strains of endophytic *Co. gloeosporioides* and *Co. fructicola* were pathogenic to citrus [10]. In contrast, similar symptoms of citrus scab have not been observed during our collection of leaves, and the species from the group Elsinoë were not found in either one of our leaf samples.

In addition, some fungi, related to the pathogens generally infecting other citrus organs, were also found in citrus leaf endophytic communities. For example, the fungi related to the pathogen of citrus fruits during storage (*Penicillium digitatum*) and related to the pathogens infecting citrus trunk, branch, and twig (species in the group of Botryospaeriaceae and Fusarium) were also found in the citrus leaf endophytic communities [12,52,57]. This phenomenon also appeared on other plants; for example, *Co. graminicola* is a causal agent of anthracnose on the phyllosphere of maize, but also colonizes the roots of maize asymptomatically [76]. These results suggest that the leaf endophytic communities may serve as a reservoir not only for leaf pathogens, but also for pathogens of other plant tissues. When their host plant is under biotic and abiotic stresses, the latent pathogens can fulfill their multiplication and infection [27,28]. However, more studies are still needed to confirm the pathogenicity of our pathogen-related fungal groups and to quantify to what extent this pathogen reservoir will contribute to disease dynamics and severity in orchards.

### 4.4. Do Fungal Pathogen-Related Groups Have Competitive Advantages in Colonizing the Leaf Endophytic Space?

For most genera (64%), we found that the OTUs assigned to pathogens and potential pathogens consisted of their solely or mainly occurring species on citrus. For example, 93.8% sequences of the genus *Colletotrichum* were from the pathogen-related groups. We infer that the virulence to citrus of some species also facilitates their endophytic colonization, thus making them more competitive than their peers at least in the same genus. Compared to closely-related nonpathogenic strains, pathogenic strains may contain virulence mechanisms and more secreted carbohydrate-active enzymes (CAZymes) which allow them to penetrate and grow into plant tissues [27,28]. For example, in the *Plectosphaerella cucumerina*-*Arabidopsis thaliana* interaction system, the content of CAZymes of one pathogenic strain is five times more abundant than that of the nonpathogenic ones [77]. After occupation of the plant inner space, the pathogens may escape the plant immunity through mechanisms like virulence gene loss and microbial interactions [27,28]. Thus, it has been thought that nonpathogenic endophytes could be disarmed pathogens [30].

Another explanation is inferred from the hypothesis that pathogens originate from nonpathogenic ancestors after gaining a virulence factor [78]. The shift of lifestyle from nonpathogenic to pathogenic can be caused by horizontal gene transfer [78,79]. In addition, through the long-term co-evolution with the host plant, many fungal endophytes have accumulated adaptive advantages in suppressing or escaping the host immunity [27,28,80]. After integration of environmental genome elements, their growth within the plant may have strengthened.

## Figures and Tables

**Figure 1 jof-10-00596-f001:**
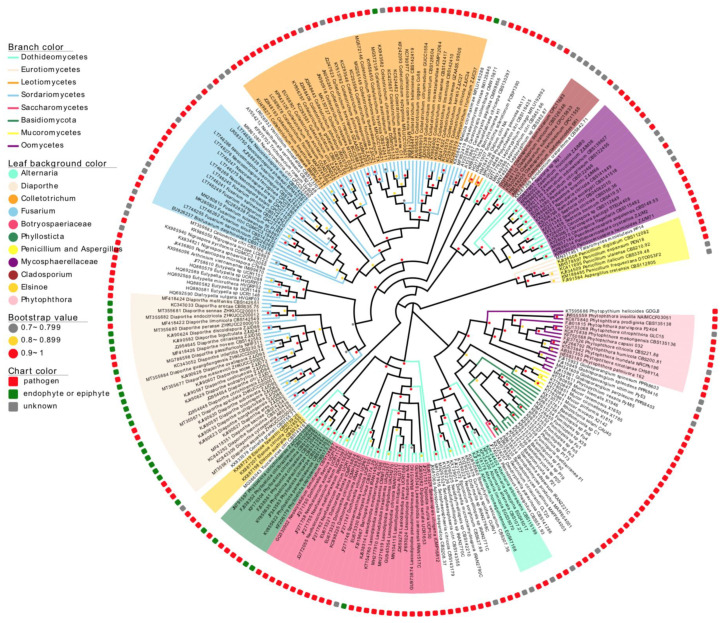
A phylogenetic tree constructed based on the ITS sequence of citrus-related fungi.

**Figure 2 jof-10-00596-f002:**
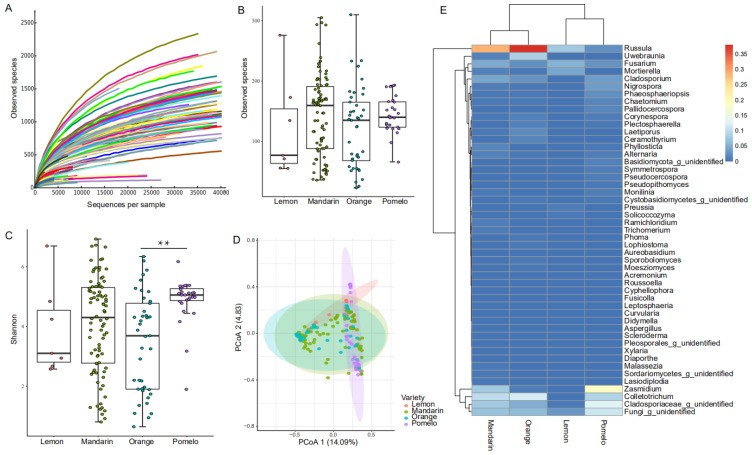
Profiling and comparison of citrus endophytic fungal communities in lemon, mandarin, orange, and pomelo. (**A**) The rarefaction curves of observed species versus the number of sequences of all 161 samples; (**B**) the observed species, and (**C**) Shannon diversity among samples of lemon, mandarin, orange, and pomelo; (**D**) principle coordinate plot of samples of, and (**E**) heatmap comparison of the relative abundance of major genera among lemon, mandarin, orange, and pomelo. The significant differences were marked with “**” (*p* < 0.01).

**Figure 3 jof-10-00596-f003:**
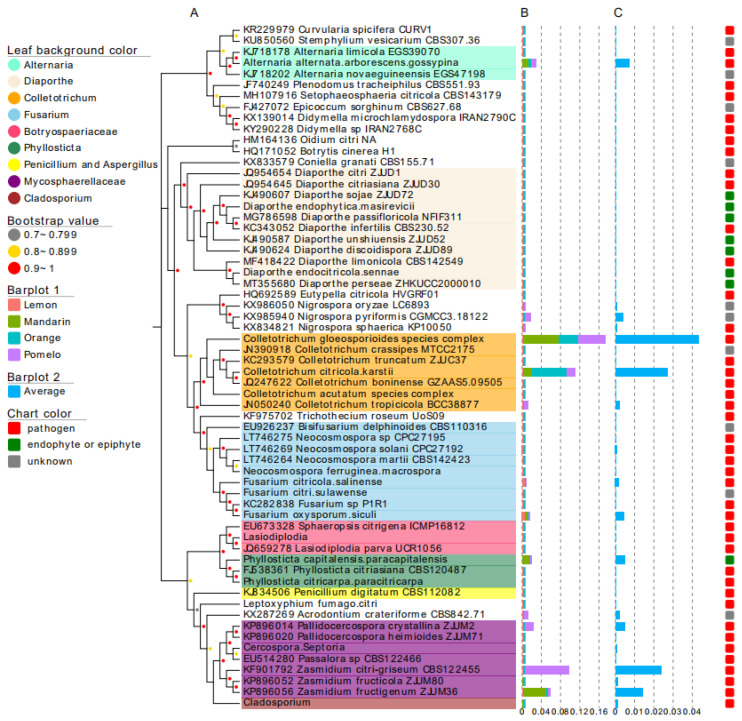
The occurrence of reported citrus-related fungi in citrus leaf endophytic communities of lemon, mandarin, orange, and pomelo. (**A**) An unrooted tree displaying 62 fungal groups occurring in citrus leaf endophytic fungal communities; (**B**) the relative abundance of the 62 fungal groups in the endophytic fungal community of lemon, mandarin, orange, and pomelo; (**C**) the average relative abundance of the 62 fungal groups in the citrus leaf endophytic fungal community.

**Figure 4 jof-10-00596-f004:**
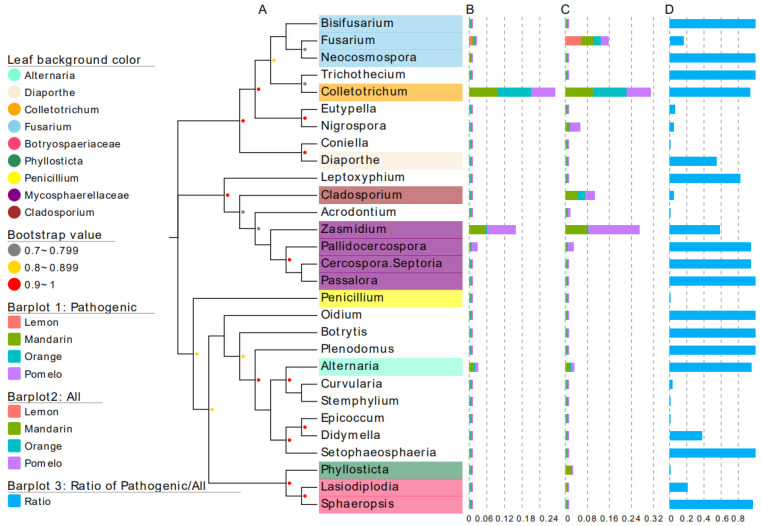
The dominance of pathogens and potential pathogens within their genus. (**A**) an unrooted tree displaying the genera occurring in citrus leaf endophytic fungal communities; (**B**) the relative abundance of OTUs of pathogens and potential pathogens in the genera listed in (**A**); (**C**) the relative abundance of all OTUS in the genera listed in (**A**); (**D**) the ratio of OTUs of pathogens and potential pathogens calculated for each genus.

**Table 1 jof-10-00596-t001:** The information of the collected leaf samples in this study.

Variety	Chongqing	Guangdong	Guangxi	Guizhou	Hubei	Jiangxi	Sichuan	Zhejiang	Sum
Lemon	0	7	0	0	0	0	0	0	7
Mandarin	8	45	8	1	4	8	0	12	86
Orange	8	0	0	0	8	17	8	0	41
Pomelo	0	20	0	0	0	0	7	0	27

**Table 2 jof-10-00596-t002:** The eleven fungal groups summarized from citrus-related fungi.

Group Name	Genera	No. of Species	Infecting Tissues	Symptoms	Influential Diseases	References
Alternaria	*Alternaria* (*A.*)	5	leaf, shoot, fruit	spot, fruit rot	Citrus brown spot, Citrus black rot	[36,37]
Botryospaeriaceae	*Barriopsis*, *Diplodia*, *Dothiorella*, *Lasiodiplodia* (*L.*), *Neofusicoccum*, *Neoscytalidium*, *Spencermartinsia*, *Sphaeropsis*	26	branch, trunk, twig	canker, dieback, gummosis	Bot gummosis	[6,12,38,39]
Cladosporium	*Cladosporium*	5	leaf	spot	-	[40,41]
Colletotrichum	*Colletotrichum* (*Co.*)	36	leaf, shoot, flower, fruit	spot, wither-tip	Citrus anthracnose, Postbloom fruit drop	[8,11,42,43,44]
Diaporthe	*Diaporthe* (*D.*)	30	leaf, fruit, twig, fruit	black points, fruit rot, dieback, gummosis	Citrus melanose, Stem-end rot	[45,46,47,48]
Elsinoë	*Elsinoë* (*E.*)	3	leaf, fruit	spot	Citrus scab	[49,50,51]
Fusarium	*Bisifusarium*, *Fusarium* (*F.*), *Neocosmospora*	17	leaf, branch, fruit, root	spot, rot, canker	-	[52,53,54]
Mycosphaerellaceae	*Cercospora*, *Pallidocercospora*, *Passalora*, *Pseudocercospora*, *Ramularia*, *Septoria*, *Zasmidium* (*Z.*)	18	leaf, fruit	spot	Citrus greasy spot	[25,55,56]
Penicillium and Aspergillus	*Aspergillus*, *Penicillium*	6	fruit	mold	Citrus green mold, Citrus blue mold	[57,58,59,60]
Phyllosticta	*Phyllosticta* (*P.*)	8	leaf, fruit	spot	Citrus black spot	[7,61,62,63]
Phytophthora	*Phytophthora*	11	root, fruit, branch	rot, canker	-	[64,65,66]

**Table 3 jof-10-00596-t003:** The species and number of OTUs included in each fungal group.

Group Name	Species	No. of OTUs
Alternaria alternata.arborescens.gossypina	*Alternaria alternata*, *A. arborescens*, *A. gossypina*	57
Cercospora.Septoria	*Cercospora* sp., *Septoria* sp.	1
Cladosporium cladosporioides.iranicum.subuliforme.tenuissimum	*Cladosporium cladosporioides*, *Cladosporium iranicum*, *Cladosporium subuliforme*, *Cladosporium tenuissimum*	21
Colletotrichum citricola.karstii	*Colletotrichum citricola*, *Co. karstii*	135
Colletotrichum gloeosporioides	*Colletotrichum aenigma*, *Co. australianum*, *Co. ciggaro*, *Co. citrimaximae*, *Co. communis*, *Co. fructicola*, *Co. gloeosporioides*, *Co. helleniense*, *Co. hystricis*, *Co. queenslandicum*, *Co. siamense*, *Co. syzygicola*, *Co. theobromicola*	384
Diaporthe citri	*Diaporthe citri*	3
Diaporthe endocitricola.sennae	*Diaporthe endocitricola*, *D. sennae*	1
Diaporthe endophytica.masirevicii	*Diaporthe endophytica*, *D. masirevicii*	1
Didymella microchlamydospora	*Didymella microchlamydospora*	3
Didymella sp IRAN2768C	*Didymella* sp.	4
Fusarium citri.sulawense	*Fusarium citri*, *F. sulawense*	2
Fusarium citricola.salinense	*F. citricola*, *F. salinense*	7
Fusarium oxysporum.siculi	*F. oxysporum*, *F. siculi*	58
Lasiodiplodia brasiliensis.caatinguensis.subglobosa	*Lasiodiplodia brasiliensis*, *L. caatinguensis*, *L. subglobosa*	1
Lasiodiplodia iraniensis.mediterranea.parva.pseudotheobromae.subglobosa.theobromae	*L. iraniensis*, *L. mediterranea*, *L. parva*, *L. pseudotheobromae*, *L. subglobosa*, *L. theobromae*	1
Leptoxyphium fumago.citri	*Leptoxyphium fumago*, *Leptoxyphium citri*	1
Neocosmospora ferruginea.macrospora	*Neocosmospora ferruginea*, *Neocosmospora macrospora*	1
Nigrospora oryzae	*Nigrospora oryzae*	5
Nigrospora pyriformis	*Nigrospora pyriformis*	5
Nigrospora sphaerica	*Nigrospora sphaerica*	5
Pallidocercospora crystallina	*Pallidocercospora crystallina*	11
Phyllosticta capitalensis.paracapitalensis	*Phyllosticta capitalensis*, *P. paracapitalensis*	1
Phyllosticta citricarpa.paracitricarpa	*P. citricarpa*, *P. paracitricarpa*	1
Zasmidium citri-griseum	*Zasmidium citri-griseum*	40
Zasmidium fructicola	*Zasmidium fructicola*	20
Zasmidium fructigenum	*Zasmidium fructigenum*	130

## Data Availability

Raw sequence data of fungal ITS amplicons were deposited under NCBI BioProject Accession PRJNA756056. The codes in analyses and plotting are available from the first author (F.H.) upon reasonable request.

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
