# Peer review of "Profiling of the Citrus Leaf Endophytic Mycobiota Reveals Abundant Pathogen-Related Fungal Groups"

_jof, 2024, doi:10.3390/jof10090596_

Round 1

Reviewer 1 Report

The authors are presenting an interesting study with broad implications and interesting results. Overall, the manuscript is well written and clear, and I would mostly suggest revising English in some sentences throughout the text where intended meaning is unclear. (Please find attached a file where sentences that require revising/editing are highlighted).

I think the introduction, while clear, should be improved with the addition of closing (final) sentences focused on a small conclusion for each paragraph (some paragraphs are clearly lacking it, e.g. first paragraph.

On the other hand, I see two weaknesses that I think the authors should address before recommending publication:

1. The idea that pathogenic fungi have competitive advantage compared to non-pathogenic congenerics when colonizing as endophytes is a very interesting observation. I think the authors should test its statistical significance (the data provided seem amenable to proper testing), and report the test. This would greatly improve their point.

2. Figure 2-E show a result that is interesting but not further discussed enough. The authors should discuss more the apparent lack of community stability apart from a bunch of genera.

Once both these comments and English revision are addressed, I think the manuscript would make a valuable contribution to the knowledge corpus of fungi community ecology.

I have highlighted in the attached file sentences or words that need revising to better convey the meaning intended by the authors.

Author Response

The authors are presenting an interesting study with broad implications and interesting results. Overall, the manuscript is well written and clear, and I would mostly suggest revising English in some sentences throughout the text where intended meaning is unclear. (Please find attached a file where sentences that require revising/editing are highlighted).

Response 1: Thank you for your comments and suggestions on our manuscript. I have tried my best to incorporate all your suggestions into the new manuscript, and hope I understood you rightly.

I think the introduction, while clear, should be improved with the addition of closing (final) sentences focused on a small conclusion for each paragraph (some paragraphs are clearly lacking it, e.g. first paragraph.

Response 2: I added a closing sentence for the first paragraph. Please check it again.

On the other hand, I see two weaknesses that I think the authors should address before recommending publication:

  1. The idea that pathogenic fungi have competitive advantage compared to non-pathogenic congenerics when colonizing as endophytes is a very interesting observation. I think the authors should test its statistical significance (the data provided seem amenable to proper testing), and report the test. This would greatly improve their point.

Response 3: I did a statistical test for this, the p values were added in the part 3.4. Please check it again.

  1. Figure 2-E show a result that is interesting but not further discussed enough. The authors should discuss more the apparent lack of community stability apart from a bunch of genera.

Response 4: The phenomenon you observed may be caused by the high presence of Russula fungi in our data. When I used log transformation, the figure looked like the one attached (Fig. 2E-log). So it’s hard to say that it lacked community stability. Please check it again.

Once both these comments and English revision are addressed, I think the manuscript would make a valuable contribution to the knowledge corpus of fungi community ecology.

Response 5: Thank you again. If I did not understand you rightly, I hope that we can get another chance to revise it.

Reviewer 2 Report

The presented study is very interesting and could be important. However, there is one big problem. Sequencing the ITS region is often insufficient for species identification, for examples, Alternaria, Fusarium, Diaporthe etc. If species are unknown, authors have no basis to discuss occurrence of pathogens, endophytes and so on. For example, many Alternaria species are pathogens, but there are also saprotrophs. Authors should distinguish which sequences represent genera and which – species. The paper should be rewritten considering level of identification.

I believe, that after re-written of paper, it will be published and bring new knowledge about mycobiota of Citrus spp.

No reasons go into the details, if main concept is wrong.

Author Response

The presented study is very interesting and could be important. However, there is one big problem. Sequencing the ITS region is often insufficient for species identification, for examples, AlternariaFusariumDiaporthe etc. If species are unknown, authors have no basis to discuss occurrence of pathogens, endophytes and so on. For example, many Alternaria species are pathogens, but there are also saprotrophs. Authors should distinguish which sequences represent genera and which – species. The paper should be rewritten considering level of identification.

I believe, that after re-written of paper, it will be published and bring new knowledge about mycobiota of Citrus spp.

Response 1: The reviewer did raise a very important question of our manuscript. It is right that a single ITS gene is not enough to assign the OTUs to fungal species. If we want to make a precise identification, we need to sequencing multi genes for each OTUs, or we need to design a specific quantitative PCR primer pairs for each pathogen. It will be a really tough work which we could not finish in a short time.

Instead of assigning our OTUs to species or genera level, and to pathogens or endophytes, we used our own criterion and assign the OTUs to “pathogen-related species”. You can understand it as the phylotypes of pathogens, e.g., the phylotype of Colletotrichum gloeosporioides, the phylotype of Alternaria alternata. This notion is still meaningful, as we put it in the Discussion, that the pathogens may evolve from closely related endophytes, and the endophytes may evolve from the pathogens with reduced virulence. These evolution trajectories were due to the long term interaction between the fungi and their citrus host.

Considering these, I hope you think our manuscript is publishable. I respect your opinion, and thank you for your reminding.

Round 2

Reviewer 2 Report

Text could be published after corrections of mistakes.

I agree with authors’ response and can accept the use of the terms “pathogen-related species” and “phylotype". Unfortunately, authors have still used the term “species” incorrectly. For example, in Figure 2 (B). and in the text: lines 177-181; line 187. I recommend carefully revising the text to avoid incorrect terminology. 

The text “caused by Colletotrichum species” should be replased with “caused by Colletotrichum spp.”, similarly in the following text.

Author Response

I agree with authors’ response and can accept the use of the terms “pathogen-related species” and “phylotype". Unfortunately, authors have still used the term “species” incorrectly. For example, in Figure 2 (B). and in the text: lines 177-181; line 187. I recommend carefully revising the text to avoid incorrect terminology. 

The text “caused by Colletotrichum species” should be replased with “caused by Colletotrichum spp.”, similarly in the following text.

Response 1: Thank you for your kindness. And also, thank you for the suggestions. I carefully checked the whole manuscript, and tried my best to avoid this kind of mistakes. However, the “Observed species” is a known term of amplicon sequencing and microbial ecology, it describes the OTU abundance in a fungal/bacterial community. So, we did not revise it.